# Genome-Wide Urea Response in Rice Genotypes Contrasting for Nitrogen Use Efficiency

**DOI:** 10.3390/ijms24076080

**Published:** 2023-03-23

**Authors:** Narendra Sharma, Dinesh Kumar Jaiswal, Supriya Kumari, Goutam Kumar Dash, Siddharth Panda, Annamalai Anandan, Nandula Raghuram

**Affiliations:** 1Centre for Sustainable Nitrogen and Nutrient Management, University School of Biotechnology, Guru Gobind Singh Indraprastha University, Sector 16C, Dwarka, New Delhi 110078, India; 2Crop Improvement Division, Indian Council of Agricultural Research (ICAR)-National Rice Research Institute (NRRI), Cuttack 753006, India; 3Institute of Agricultural Sciences, SOA (DU), Bhubaneswar 751003, India; 4Regional Station, Indian Council of Agricultural Research (ICAR)-Indian Institute of Seed Science, Bengaluru 560065, India

**Keywords:** nitrogen use efficiency, urea, networks, QTLs, rice, transcriptome

## Abstract

Rice is an ideal crop for improvement of nitrogen use efficiency (NUE), especially with urea, its predominant fertilizer. There is a paucity of studies on rice genotypes contrasting for NUE. We compared low urea-responsive transcriptomes of contrasting rice genotypes, namely Nidhi (low NUE) and Panvel1 (high NUE). Transcriptomes of whole plants grown with media containing normal (15 mM) and low urea (1.5 mM) revealed 1497 and 2819 differentially expressed genes (DEGs) in Nidhi and Panvel1, respectively, of which 271 were common. Though 1226 DEGs were genotype-specific in Nidhi and 2548 in Panvel1, there was far higher commonality in underlying processes. High NUE is associated with the urea-responsive regulation of other nutrient transporters, miRNAs, transcription factors (TFs) and better photosynthesis, water use efficiency and post-translational modifications. Many of their genes co-localized to NUE-QTLs on chromosomes 1, 3 and 9. A field evaluation under different doses of urea revealed better agronomic performance including grain yield, transport/uptake efficiencies and NUE of Panvel1. Comparison of our urea-based transcriptomes with our previous nitrate-based transcriptomes revealed many common processes despite large differences in their expression profiles. Our model proposes that differential involvement of transporters and TFs, among others, contributes to better urea uptake, translocation, utilization, flower development and yield for high NUE.

## 1. Introduction

Sustainable nitrogen (N) management is crucial for sustainable food systems and mandated by two UN resolutions (UNEP/EA.4/Res.14 and UNEP/EA.5/Res.2) to prevent N-pollution and its impacts on health, biodiversity and climate change [1,2,3,4]. Plant biologists have a major role in improving crop nitrogen use efficiency (NUE) to minimize fertilizer wastage and pollution [5,6,7].

Rice is among the most produced and consumed crops globally, with the lowest NUE [8] and highest N-fertilizer consumption [9], making it an ideal and post-genomic crop to improve NUE. Urea is the predominant N-fertilizer used in rice-growing in developing countries, whereas ammonium nitrate is preferred in the developed world. This necessitates biological understanding of NUE in a N-form specific manner, as was carried out while we were phenotyping for NUE [10,11]. Many QTLs have been identified in rice mainly for N-response and partly for NUE, though dissecting the associated genes is in progress [12,13,14,15].

Functional genomics revealed various N-responsive genes/processes in rice (reviewed in [14,16,17]). Most of them were nitrate-based and only one each was based on urea [18] or thiourea-based (GSE71492). Few studies have undertaken a systematic shortlisting of NUE candidate genes/processes from transcriptomes associating genes with QTLs, phenotype, etc. [14]. Nevertheless, some successful examples of NUE improvement in rice included targeted assimilation, transporters [19,20,21,22,23,24,25], transcription factors (TFs) [26,27,28,29,30,31,32,33] and others [12,29,34,35,36,37,38,39].

Comparative transcriptomic studies focused on nitrate response in genotypes with contrasting yield [13,40] or NUE in rice [16], but such studies are lacking for urea, except one field study described by Neerja et al. [41]. In view of the difficulties associated with interpreting the effects of dynamic mixtures of N-forms in field soils, we report here the first controlled study of urea response in contrasting rice genotypes that were field-validated for NUE.

## 2. Results

### 2.1. Transcriptomic Analyses of Rice Genotypes with Contrasting NUE under Low Urea

To understand the genome-wide effects of urea, two indica rice genotypes Nidhi (low NUE) and Panvel1 (high NUE), were grown for 21 days in nutrient-depleted soil containing normal (15 mM) and low (1.5 mM) urea and used for microarray analyses, as described earlier [10,11,16]. The raw data were deposited in NCBI GEO (GSE140257). Scatter plots revealed good correlations (R^2^) between two of the three independent replicates (Appendix A), which were selected for further analyses. Differentially expressed genes (DEGs) were identified using geometric mean fold change value ±1.0 (log_2_FC) with a statistically significant cut off (*p* value ≤ 0.05). Volcano plots revealed the higher number of DEGs in Panvel1 (2819) as compared to Nidhi (1497) under low urea (Figure 1A,B). There were 704 up-regulated and 793 down-regulated DEGs in Nidhi, whereas 1241 up-regulated and 1578 were down-regulated DEGs in Panvel1 (Figure 1C). The DEGs common between both the genotypes were 271, of which 34 DEGs were oppositely regulated in both the genotypes (Appendix A).

### 2.2. Contrasting Rice Genotypes Reveal Common and Distinct Processes in Response to Urea

Our search for biological processes specific to the genotypes in low urea revealed a total of 353 total Gene Ontology (GO) terms for Nidhi and 493 for Panvel1. They include photosynthesis, light harvesting in photosystem I, response to light stimulus, protein-chromophore linkage, photorespiration, reductive pentose-phosphate cycle, response to water deprivation, water transport and response to cold; these biological processes are common and enriched in the urea response of both of the genotypes (Figure 1D, Appendix A). Microtubule-based process, mitotic cell cycle, chromatin organization, response to oxidative stress, protein glutathionylation and alternative respiration, among others, were highly enriched only in Nidhi under low-urea conditions (Appendix A). However, biological processes related to protein oligomerization, response to heat, response to reactive oxygen species, glutathione metabolic process, cold acclimation, UDP-glucosylation and carbon fixation, among others, were predominant only in Panvel1 (Appendix A).

### 2.3. RT-qPCR Validates Urea-Regulation of DEGs from Selected Processes

Eight genes were selected based on their involvement in nitrogen transport, starch synthesis, photosynthesis and flowering time. Their urea-regulated gene expressions observed in microarray data were validated by RT-qPCR under normal urea (15 mM) and low-urea (1.5 mM) treatments in both the genotypes contrasting for NUE (Figure 2). Among them, four DEGs codes for Photosystem I reaction center subunit N (*psaN*, Os12g0189400), UDP-Glucose-dependent glycosyl transferase 703A2 (*UGT703A2*: Os01g0638000), days to heading on chromosome 2 (*DTH2*, Os06g0298200) and light-harvesting protein ASCAB9-A (*ASCAB-9A*, Os11g0242800) were common to both of the genotypes. Except *DTH2*, all of them were down-regulated by low urea in both of the genotypes. Among the DEGs exclusive to high-NUE genotype Panvel1, three down-regulated DEGs were validated, viz., pseudo-response regulator 2 (*PRR95*, Os09g0532400), starch synthase-IIb (*OsSIIb*, Os02g0744700) and Nuclear Factor-Y subunit B8 (*NF-YB*, Os03g0413000). Among the DEGs exclusive to the low-NUE genotype Nidhi, an up-regulated DEG encoding a voltage-dependent anion channel 6 (*OsVDAC6*, Os03g0137500) was validated. The list of primers used in this study is provided in Appendix A.

### 2.4. Transporters Respond Differentially to Urea in Contrasting Rice Genotypes

Nitrate transporters have been reported to regulate source-sink flux and NUE [14,14,37,42]. However, urea regulation of genes encoding transporters or their roles in source-sink fluxes and/or NUE are not well understood to the best of our knowledge. In this study, mining of DEGs in a transporters’ database provided 63 urea-responsive transporters belonging to 21 families for low-NUE genotype Nidhi, with nearly equal proportion of up- and down-regulated DEGs (Table 1 and Appendix A). The high-NUE genotype Panvel1 revealed 107 urea-responsive transporters belonging to 36 families with more up-regulated than down-regulated transporters (Appendix A). Venn selections revealed 15 transporters common to both genotypes belonging to 10 distinct families, whereas 48 transporters were exclusive to Nidhi and 92 were exclusive to Panvel1. This clearly shows that contrasting genotypes responded differentially to urea in regulating various transporters (Appendix A).

Further Venn selections with transporters previously linked to NUE in rice [14] revealed 5 transporters, viz., *OsHKT8, OsPUP4, OsPUP7, NRAMP5* and *Lsi3* as related to NUE in the genotype Nidhi. Further, 24 of the 63 transporters identified in Nidhi were associated with other physiological traits, viz., abiotic stress, root and other various functions, etc., whereas 35 others are completely novel and functionally unvalidated (Appendix A). For example, Os03g0823500 is a novel member found to be urea-responsive from the PTR family, though another member of the same family, *OsPTR9*, has been reported to be involved in nitrogen metabolism [20]. Similarly, we found *OsSultr2;2* as a novel urea-responsive member of sulphate transporters’ family, from which OsSultr3;3 was shown to alter the metabolite profile in rice grains. We also found *OsABCG7*, a novel urea-responsive member of the ABC transporter family, of which *OsABCG26* is known for its role in reproduction process [43], and *OsABCG3* in pollen development [44].

Similar analysis for Panvel1 revealed 8 transporters related to NUE, viz., *OsFRDL1, MIT, OsHKT7, OsMATE2, OsNramp3, OsABCG16, NPF7.1* and *OsZIP4* [14], of which only *NPF7.1* is nitrate-responsive. Thirty other transporters are related to abiotic stress, micro and macro nutrient regulation, water transport and development, among others. Further, 67 other transporters are completely novel and functionally unvalidated (Appendix A). For example, *OsLHT9* is the novel candidate found to be urea-responsive from the LHT family; however, knockout rice plants for *OsLHT1*, a member of the same family, showed reduced N-uptake and utilization efficiency [45]. Similarly, *NRT1.1C* is the novel member found, as urea responsive belongs to the NRT family, while its other member *NRT1.1A* has been reported for its role in seed maturation and yield [42]. We also found *OsCOPT6* a novel urea-responsive candidate of copper transporters family, though *OsCOPT1* and *OsCOPT5* have been reported for their role in copper metabolism and virus resistance [46].

Thus, we identified 13 transporters as differentially regulated by urea in contrasting rice genotypes for the first time (5 in Nidhi and 8 in Panvel1), apart from 102 novel transporters (35 + 67) (Appendix A). Further characterization of these 121 urea-regulated, NUE-related transporters should enable the shortlisting of candidates to improve N/urea use efficiency in rice. Considering that barring *NPF7.1*, none of them is known to transport any N-form, it clearly means that NUE involves N-regulation of a large number of non-N transporters. Further, *NPF7.1* seems to be an important NUE candidate, as it responds to both nitrate and urea and is also associated with yield and therefore NUE.

### 2.5. Different Transcription Factors Mediate Urea Response in Contrasting Rice Genotypes

TFs mediate plant responses to external signals by regulating the expression of the genes involved. By searching for urea-responsive DEGs in the PlantPAN3 database, we identified 45 TFs belonging to 18 TF classes in Nidhi, and 96 TFs belonging to 28 classes in Panvel1 (Table 1 and Appendix A). In Nidhi, all the members of bZIP and Homeodomain TF families were up-regulated, while the members of AP2, C2H2, C3H and ERF TF families were down-regulated under low urea. Further, there were an equal number of up- and down-regulated TFs of NAC-NAM and WRKY families. In Panvel1, the members of TF family C2H2, B3, SBP, TCR, MADS and YABBY were up-regulated. Members of AP2, AT-Hook, bHLH, bZIP, ERF, homeodomain, MYB, NAC/NAM and WRKY families have more up-regulated TFs than down-regulated, while most of the HSF TF family members were more down-regulated than up-regulated.

To identify NUE-related TFs among these urea-regulated TFs, Venn selections were carried out with 237 NUE-related TFs we reported earlier [14]. It revealed 13 NUE-related TFs differentially regulated by urea in Nidhi, while 20 were associated with biotic/abiotic stress, development and senescence, including others, whereas 12 are completely novel and functionally unvalidated (Appendix A). For example, *WRKY37* is a novel urea-responsive TF belonging to WRKY family, though the roles of other WRKY TFs have been reported, such as in abscisic acid signaling and auxin transport [47] and UV tolerance [48]. Similarly, *OsGRF2* is a novel member found to be urea responsive from the GRF family. Interestingly, the over expression of another member, *OsGRF4* has been reported to regulate grain shape and panicle length [49].

Similarly, in Panvel1, 27 NUE-related TFs were found to be urea responsive [14], while 37 TFs were linked with other traits such as abiotic/biotic stress, root, leaf, hormone, germination and phosphate regulation, etc. In addition, 32 other TFs found to be urea responsive in our study are completely novel and functionally unvalidated (Appendix A). For example, *LBD11-1* is a urea-responsive novel candidate from LOB family, while another candidate of same family *LBD12-1* has been reported for stunted growth, twisted leaves and abnormal anthers [50]. Similarly, we found *OsbHLH31* as a novel urea-responsive member of bHLH transcription factor’s family, from which *OsbHLH35* was shown to link with another development [51]. Therefore, 40 urea-regulated and NUE-related TFs (13 in Nidhi and 27 in Panvel1) should enable the shortlisting of candidates for improving N/urea use efficiency in rice, as many of them are common to urea and other N-forms.

To further characterize the differential deployment of urea-responsive TFs in the contrasting genotypes Nidhi and Panvel1, TF binding sites (TFBS) were predicted/searched in the promoter regions of all DEGs using Regulatory Sequence Analysis Tools (RSAT) (http://plants.rsat.eu accessed on 1 September 2021). A majority of 24 DEGs in Nidhi revealed TFBS as belonging to the TF family AP2EREBP, followed by 5 DEGs for NAC and 3 DEGs for bZip families, among others (Appendix A). GO enrichment analysis revealed that they are involved in translation, embryonic development ending in seed dormancy, response to cadmium ion, mismatch repair, mitochondrial transport, etc. Further, the molecular functions of genes containing these TFBS include structural constituent of ribosome, ATP-dependent helicase activity and metal ion transmembrane transporter activity in Nidhi (Appendix A). These are novel urea-responsive cis regulatory elements (CREs) unknown in any crop and therefore merit further characterization.

Similarly, in Panvel1, these TFBS are involved in protein amino acid phosphorylation, ATP binding, translation, mitochondria, embryonic development ending in seed dormancy, xanthophyll biosynthesis, mitochondrial transport, etc. Their functions include protein serine/threonine kinase activity, ATP-dependent helicase activity, RNA binding CC chloroplast stroma, ATP binding, etc. (Appendix A).

### 2.6. Different miRNAs Post-Transcriptionally Regulate Urea Response in Contrasting Genotypes

Post-transcriptional regulation by miRNAs is implicated in many important traits in rice development, growth, stress tolerance [52,53] and yield [54], including NUE [14,16]. However, they have not been characterized for urea response in NUE. Our search for miRNA targets among urea-responsive DEGs using the Plant miRNA database (http://bioinformatics.cau.edu.cn/PMRD/) (accessed on 05/08/2021) revealed 45 DEGs in the low-NUE genotype Nidhi with nearly equal proportion of up-/down-regulated genes (Table 1 and Appendix A). A higher number of miRNA targets (57) were found in the high-NUE genotype Panvel1, with more up-regulated than down-regulated genes as potential miRNA targets (Appendix A).

Venn selections between genotypes revealed six common miRNAs, viz., osa-miRf10059-akr, osa-miR806a, osa-miRf10549-akr, osa-miR2092-5p, osa-miRf10141-akr and osa-miRf10531-akr. In order to check whether these miRNAs could significantly contribute to NUE, Venn analyses of these miRNAs was performed with 69 our recently predicted miRNA targets for NUE in rice [14]. It revealed 3 NUE miRNAs osa-miR1436, osa-miR1861c and osa-miR1863 only in Nidhi, while none were found in Panvel1. Thus, we identified 42 miRNAs as novel candidates in Nidhi and 57 in Panvel1 (Appendix A) to be validated further for their role in improving NUE in rice. Genotype-wise Venn analyses of these miRNA targets with the 69 we identified earlier as NUE-related in rice [14] yielded 3 NUE-related urea-responsive miRNAs: osa-miR1436, osa-miR1861c and osa-miR1863, only in Nidhi, while none were found in Panvel1.

GO analysis of DEGs in Nidhi using ExPath 2.0 revealed their role in carbohydrate transport, metabolic process, phosphorylation and transmembrane receptor protein serine/threonine kinase signaling (Appendix A). GO of DEGs in Panvel1 showed processes such as cytoplasmic translational initiation, formation of translation initiation complex xylan biosynthetic process, phosphorylation, microtubule-based movement, translational initiation intracellular signal transduction and dephosphorylation. The details of their genes and functions are provided in Appendix A. These results indicate differential regulation of miRNA targets in contrasting rice genotypes and their involvement in the urea regulation of translational, post-translational and carbohydrate metabolism processes, among others.

### 2.7. PPI Networks Reveal Differential Urea Responsive Interactions in Contrasting Genotypes

To construct the protein-protein interaction (PPI) networks, DEG-associated interactors detected experimentally were retrieved from STING, PRIN, MCDRP and BioGRID databases, as described earlier [16]. The interactors were used to construct PPI networks for each genotype and the expression values of the DEGs were mapped onto the networks in Cytoscape (Appendix A). The networks consisted of 831 nodes and 2583 edges in Nidhi, whereas 999 nodes and 3591 edges were detected in Panvel1. Venn analyses using interactors detected in both of the networks revealed 317 interactors common to urea response in both genotypes, whereas 518 and 683 were exclusive to Nidhi and Panvel1, respectively (Appendix A). Their functional annotation revealed that oxidation-reduction, isoprenoid and carotene biosynthesis, among others, were uniquely enriched in Nidhi, whereas flower development, ovule development and regulation of meristem development were uniquely enriched in Panvel1 (Appendix A).

The complex PPI networks were sub-clustered into smaller networks (molecular complexes) using MCODE plugin in Cytoscape. In Nidhi, 11 subclusters were detected, whereas 21 subclusters were detected in Panvel1 (Appendix A). When molecular complexes with a MCODE score >4 were considered for further analyses (Appendix A), the top scorer consisted of 29 nodes and 376 edges in Nidhi, and 23 nodes and 109 edges in Panvel1. Functional annotation of genes associated with these top clusters revealed their involvement in protein translation in Nidhi and also embryo sac development in Panvel1 (Appendix A). Another interesting observation in the PPI networks was that the D1 node interacts with many proteins in both genotypes (Appendix A). D1 is a natural mutant of japonica rice devoid of functional G-protein alpha subunit (RGA1). Our recent analysis of its transcriptome indicated G-protein regulation of N-response. The finding that RGA1 constitutes an important node in urea response in the present study adds mechanistic insights into the underlying protein interactions.

### 2.8. Urea NUE Involves Differential Post-Translational Regulation in Contrasting Genotypes

As post-translational modifications (PTMs) emerged in the GO analysis of DEGs, their details were analyzed using PTM viewer (https://www.psb.ugent.be/webtools/ptm-viewer/experiment.php accessed on 1 September 2021). Of the 552 DEGs encoding proteins with potential PTMs in the low-NUE genotype Nidhi, phosphorylation was predominant (240), followed by lysine 2-hydroxyisobutyrylation (224), lysine acetylation (52), carbonylation (18), N-glycosylation (10), ubiquitination (7) and succinylation (1). The high-NUE genotype Panvel1 revealed many more (852) DEGs with more extensive PTMs of their proteins in similar order, with phosphorylation (436), hydroxy isobutyrylation (282), acetylation (96) and carbonylation (27). However, it had lower ubiquitination (6) and N-glycosylation (4), and similar succinylation (1). Interestingly, only 118 DEGs were common to both the genotypes, clearly indicating that NUE involves targeting different proteins for PTMs in contrasting genotypes (Appendix A). Further comparison by the type of PTMs revealed that most of the differential protein targets involved lysine 2-hydroxyisobutyrylation (59) followed by phosphorylation (49), lysine acetylation (9), carbonylation (7) and lysine ubiquitination (1). These results highlight the hitherto unknown importance of PTMs in general and lysine 2-hydroxyisobutyrylation and phosphorylation in particular in urea response and NUE, which can only be identified by studying contrasting NUE genotypes.

Transporters and TFs are the two major functional categories in which PTMs could play crucial roles in NUE. To test this hypothesis, the 1404 urea-responsive PTM targets identified here were searched among 85 transporters and 237 TFs we identified previously as NUE-related [14]. It revealed 11 PTM targets as NUE-related in Nidhi, including 10 TFs and one transporter (Appendix A). Among them, 8 TFs, viz., OsPCL1, OsPRR73, RF2B, OsMyb4, OsRI, OSH15, qHD2(t) and Hd18 and a transporter (OsPUP7) were phosphorylated, while 2 other TFs, OsC3H33 and OsJMJ706 were acetylated. Similarly, the high-NUE genotype Panvel1 revealed 18 TFs and two transporters as NUE-related. Sixteen of these 18 TFs were phosphorylated, viz., OsJAZ2, DOS, OsKn2, SPL6, OsIDD2, SPL16, OsSCR, YAB2, NF-YB8, OsPRR95, NF-YB10, OsTGA2, OsMYB1R1, OsMyb4, qHD2(t) and Hd18, while 2 other TFs (OsJAZ4 and OsWRKY11) were acetylated. Both the NUE-related transporters (OsHKT7 and OsNramp3) were modified by phosphorylation.

Thus, our analysis identified 1404 urea-responsive PTM targets (552 in Nidhi and 852 in Panvel1) and shortlisted 31 of them as NUE-related for the first time (11 in Nidhi and 20 in Panvel1). Further, we narrowed down 2 out of 7 types of urea-responsive PTMs as NUE-related, dominated by phosphorylation followed by acetylation. Such experimental distinction of N-response and NUE at the level of PTMs was previously unknown and could be of strategic significance for crop improvement.

### 2.9. Contrasting Genotypes Reveal Different NUE Candidates by QTL Co-Localization

Yield-association has been our most important differentiator between N-response and NUE [11,14]. Venn selection using 1497 urea-responsive genes identified here in Nidhi and an updated list of 3532 yield-related genes revealed 252 N-responsive and yield-related genes, which we termed urea NUE-genes (Figure 3A). Similar Venn selection of 2819 urea-responsive genes in Panvel1 with 3532 yield-related genes identified 317 urea NUE genes (Figure 3B). These 252 and 317 urea NUE genes were co-localized onto NUE-QTLs updated from [14]. This revealed 69 unique urea NUE genes colocalized to 31 NUE-QTLs in Nidhi. They were mostly on chromosomes 1 (20), 3 (18) and 9 (9), followed by 6 each on chromosomes 5 and 7, 3 each on chromosomes 6, 11 and 12 and one on Chromosome 4 (Figure 3C–E, Appendix A). Process annotation of these 69 urea NUE-candidates revealed carotenoid biosynthesis, starch and sucrose metabolism as the most enriched pathways in Nidhi (Appendix A).

This revealed 96 unique urea NUE genes colocalized to 21 NUE-QTLs in Panvel1. They were mostly on chromosomes 3 (28), 1 (22) and 9 (16), followed by 10 genes on chromosome 5, 6 (7), 7 (5) and 4 genes on chromosome 3 (Figure 3F–H, Appendix A). Process annotation of these 96 urea NUE candidates in Panvel1 revealed circardian rhythm, photosynthetic carbon fixation and metabolism of cyanoamino acid, nitrogen porphyrin and chlorophyll as the most enriched pathways (Appendix A).

### 2.10. Low Urea Enhances Photosynthetic, Transpiration and Water Use Efficiencies in High-NUE Genotype

To experimentally test any role for photosynthetic efficiency, transpiration efficiency and internal water efficiency in NUE, 21-days-old potted plants of Nidhi and Panvel1 were monitored using LICOR6400XT as described in materials and methods. All three efficiencies were significantly higher (*p* < 0.05) in low urea over normal urea for the high-NUE genotype Panvel1. Only photosynthetic efficiency (but not other efficiencies) was only slightly higher in low urea in Nidhi (Figure 4), indicating that better photosynthetic performance of Panvel1 under low urea may contribute to its better NUE.

### 2.11. Field Data Validate NUE in Two Genotypes

The field performance of the two rice genotypes with contrasting NUE used in this study (Nidhi and Panvel1) was evaluated with no added N (N0), 50 kg/ha N (N50) and 100 kg/ha N (N100) supplied as urea. The yield and NUE parameters, viz., yield per hectare (kg/ha), 1000 grain weight (gram), panicle weight (gram), yield per plant (gram), uptake efficiency, partial factor productivity, nitrogen transport efficiency, utilization efficiency and fertilizer use efficiency were measured at maturity stage. The high-NUE genotype Panvel1 performed significantly better in all these parameters, relative to the low- NUE genotype Nidhi (Figure 5).

### 2.12. Common and Distinct Effects of Nitrate and Urea on NUE in Contrasting Genotypes

Taking advantage of our simultaneous studies for nitrate and urea response using the same contrasting genotypes, our current urea-responsive microarray data were compared with the corresponding nitrate-responsive data [16] (Figure 6). A higher number of 1397 DEGs responded to low nitrate in Nidhi more than the 735 DEGs in Panvel1 (Figure 6A), whereas in low urea, Panvel1 displayed a higher number of 2819 DEGs than the 1497 in Nidhi (Figure 6A). Venn selection of nitrate and urea for each genotype revealed fewer common DEGs in Nidhi (115) than in Panvel1 (145), suggesting that majority of the genes are unique to either nitrate or urea response in each genotype (Figure 6A). Venn analyses of DEGs assigned to the various pathways (including non-significantly enriched) revealed that a majority of these pathways are common, clearly indicating that nitrate and urea regulate N-response through the same pathways using different genes (Figure 6B, Appendix A).

Combining the N-response data for both the genotypes revealed a total of 1530 nitrate-responsive genes and 3513 urea-responsive genes (Figure 6C,D). Pathway annotation of these nitrate-responsive genes revealed their involvement in pyruvate-, thiamine, betalain- and histidine-metabolism (Figure 6C, Appendix A), Similar analysis of urea-responsive genes revealed they are involved in different pathways, viz., photosynthesis, glyoxylate and dicarboxylate- and glutathione-metabolism (Figure 6D, Appendix A). These data suggest that urea regulates carbon metabolism and associated signaling more prominently than nitrate, to modulate crop yield/NUE under low N condition. The 532 DEGs common to nitrate and urea response displayed genotype-specific patterns of expression as shown in the heatmap (Figure 6E). Evaluation of nitrate- and urea-dependent common genes revealed co-regulated, opposite and mixed expression patterns, including variation in their extent of regulation for many common genes (Figure 6E).

To delineate nitrate and urea responses at the molecular level, the top 10 enriched transporter families were compared, which revealed the common regulation of AAAP, ABC, MFS and POT, among others, under low-nitrate and urea conditions (Figure 6F). Enriched nitrate-regulated transporter families were Amt, ClC, GIC and GPH, etc., whereas urea-regulated transporter families were ATPase, IISP and MC, among others (Figure 6F). A comparison of the top 10 enriched TF families showed AP2, bHLH, bZIP, NAC and WRKY families were common to nitrate and urea response, even if individual genes are differentially regulated (Figure 6G). The TCP, MIKC and ARF, among others, TF families were nitrate regulated, whereas AT-Hook, ERF, and GRF were regulated by urea (Appendix A).

## 3. Discussion

The roadmap for crop improvement towards NUE is still evolving, especially in biological terms [4,7]. This is mainly due to the many different definitions of NUE [6] and inadequate experimental differentiation between N-response and NUE, especially with respect to different N-forms/doses in fertilizers and in field soils. The development of N-form and dose-specific media and nutrient-free soil aided in precise control on experimental conditions in rice [10,55]. They facilitated the recent characterization of the NUE phenotype and identification of contrasting genotypes validated in the field [10,11], as well as transcriptomic identification of N-responsive genes/processes [17,56]. Using yield to differentiate between N-response and NUE aided in the shortlisting of important candidate genes/processes/QTLs for NUE in rice [11,14], including from nitrate response in contrasting genotypes [16]. The current study builds on these findings to specifically address genome-wide urea response and NUE in the same two contrasting indica rice genotypes (Panvel1 for high NUE and Nidhi for low NUE) for the first time. We explain their differences in NUE in terms of some common but largely differential involvement of genes/processes involving nitrogen transport, TFs, miRNAs, post-translational modifications and QTLs. We also demonstrate differences in nitrate and urea responses and offer some candidates for crop improvement, especially since urea is the most predominant fertilizer for rice globally and NUE improvement in rice effectively means urea use efficiency.

Our transcriptomic analysis compared low urea response relative to normal urea in potted plants grown on nutrient-free sterilized soil. There were significant differences in their response to low urea, with Panvel1 showing nearly twice the number of DEGs than Nidhi, with proportionately high up-regulated DEGs (Figure 1, Appendix A). This clearly indicates that high NUE may involve many more DEGs than low-NUE in urea response, whereas the opposite was true in nitrate response for the same rice genotypes and conditions [16]. While these differences have been explored in detail (see later), it is important to emphasize two important findings here that were not reported in any of the transcriptomic studies on nitrate and/or urea response: A) that it requires analysis of multiple (ideally contrasting) genotypes under identical conditions to discern these differences and B) all functional biological interpretations of NUE or attempts for its improvement must be made in terms of specific N-forms, or assumed to vary with other N-forms unless ruled out experimentally.

Photosynthesis is one of the most important processes in crop N response and NUE [57] and its proteins are among those prominently affected by nitrogen status [34,58]. Our RT-qPCR data validate the higher downregulation of the DEG *PsaN* (photosystem I subunit N) by low urea in the low-NUE genotype relative to the high-NUE genotype. These findings are consistent with similar results in wheat regarding a gene encoding photosystem II 10 kDa protein in wheat [59]. The expression of *PsaN* is also consistent with our physiological data, showing less photosynthetic efficiency in low-NUE genotype than in high-NUE genotype (Figure 4). Further, our data on higher photosynthetic efficiency, transpiration and internal water use efficiency in Panvel1 over Nidhi (Figure 4) confirm our transcriptomic findings on the involvement of these processes in urea response/NUE. They seem to be independent of N-form, as they were also reported in nitrate response/NUE in rice [14,16,17].

Among the 34 urea-responsive DEGs that were common to both the genotypes but were oppositely regulated, there was Zink finger protein 15 (*ZFP15*). It is associated with plastid/ion binding activity (Os03g0820400, GO:0009536) and was predicted to be involved in NUE in rice [14]. In our present study, its upregulation in the high-NUE genotype Panvel1 relative to the low-NUE genotype confirms its predicted involvement in NUE and makes it an attractive candidate for crop improvement. Over-expressed dehydration-responsive element binding1A (*DREB1A*) has been shown to improve drought tolerance in rice [60]. In our microarray data, it was up-regulated in the high-NUE genotype Panvel1 while downregulated in Nidhi, which makes it an important candidate to improve drought tolerance as well as NUE. Another overexpressed gene, *OsSIDP366*, led to enhanced drought and salt tolerance in rice [61], which we found to be up-regulated in the high-NUE genotype Panvel1, while it was down-regulated in Nidhi. To the best of our knowledge, these genes constitute the first evidence of convergence among candidate genes to improve NUE as well as abiotic stress tolerance, which is of prime agronomic significance for the farmers.

Transporters and TFs are emerging as key regulators of N-response/NUE [14,17,32,56]. Transporters exist as families for various metabolites and nutrients, including for nitrate, ammonium and urea. N-transporters are crucial for N uptake efficiency and some of them such as *AMT1.1, NRT1.1A, NRT1.1B, NRT2.3b, OsNPF7. 7*, *OsNPF7.9* [25] and *OsAAP1* [62] have been validated for NUE in rice. However, there are no such studies to our knowledge on urea transporters or other transporters regulated by urea in NUE in any crop. Our N-form specific study on urea response in nutrient-depleted soil devoid of urease activity clearly shows that urea regulation of NUE goes beyond N-transporters found in nitrate-based studies. It is interesting that out of the 13 urea-responsive NUE-related transporters we found in contrasting rice genotypes, one is a peptide transporter and others predominantly deal with other nutrients such as K, P, Zn, etc. (Appendix A). Such results were also found in the SNP haplotyping of Aus landraces under low nitrogen status of rice seedlings for root vigor [63]. This means urea mediates N-regulation of other transporters towards nutrient homeostasis that may be essential for NUE, thus contributing to overall nutrient use efficiency. This is hitherto unknown in any crop and is worth pursuing further towards crop improvement.

TFs are not characterized in urea response but are well known in nitrate response/NUE [5,17], though attempts to find nitrate response elements as binding sites for TFs have not borne fruit [56]. However, 7 N-responsive TFs have been validated for their role in NUE in rice [27,29]. In our current study, of the 128 urea-responsive TFs, 15 are known to be nitrate-responsive [14], while the rest are novel and merit further evaluation for their role in NUE.

The role of PTMs in NUE has largely been ignored, as gene expression and protein expression dominate the literature. However, their roles in N-response, utilization and signal transduction are known [64]. Our study revealed 7 types of urea-responsive PTMs in 1286 proteins, mostly phosphorylation followed by acetylation, though ubiquitination was also found, which was linked to source-to-sink nitrate remobilization in Arabidopsis [65]. More importantly, we show for the first time that high NUE is associated with more extensive PTMs in hundreds of more proteins in Panvel1 than in the low-NUE genotype. It is possible that there are many other N-regulated PTMs of proteins whose genes are not differentially expressed in an N-responsive manner and are therefore not discernible by transcriptomics. Even proteomic comparisons of differential protein expression do not automatically reveal PTMs, unless specifically investigated for N-responsive PTMs. Future efforts to reconcile transcriptomes and proteomes for N-response/NUE must pay special attention to PTMs, as they may change the intervention strategies for crop improvement.

Many QTLs for NUE have been reported in rice and the genetic dissection of associated genes is in progress [14,15,66]. Our effort to co-localize urea-responsive and yield-related genes onto reported NUE-QTLs revealed that Chromosomes 1, 3 and 9 contain most of the candidate genes, adding 137 novel candidates to those reported by Kumari et al. [14]. Such chromosomal hotspots are of great value for plant breeders to improve NUE. Further, our process annotations of the genes in these chromosomal hotspots revealed their involvement in N/C fixation, porphyrin and chlorophyll metabolism, etc., in Panvel1 and carotenoid, carbon and sucrose metabolism in the low-NUE genotype.

Our comparison of the present genome-wide urea response data with our previous nitrate response data generated in the same genotypes under identical conditions provides three important insights that can only be obtained in such comparisons. First, among the well-known common DEGs between nitrate and urea, a urea transporter (*OsDUR3*) was down-regulated by low nitrate but up-regulated by low-urea conditions. Second, among transporters, ammonium transporters were regulated primarily by nitrate, whereas ATP-dependent ion transporters were regulated by urea, clearly indicating mutual regulation of the uptake/transport of different N-forms that commonly exist in field soils. Third, TFs families such as TCP were nitrate-responsive, while AT-hook TF was urea-responsive (Appendix A). It has been shown that AT-hook TF interacts with TCP TF [67], which was involved in the regulation of nitrate response in rice [68]. While their role in rice NUE needs validation, our results indicate that their regulation by different N-forms may play a role in it as a means of balancing the plant’s needs in response to variable mixtures of different N-forms in field soils.

Our analysis of the PPI networks revealed the enrichment of flower development, ovule development, embryo sac development and regulation of meristem development in the higher NUE genotype Panvel1 (Appendix A). These processes are consistent with our earlier delineation of the phenotype for urea response and NUE [11], which identified flowering time and root/shoot length as important phenotypic traits for NUE. We also found G-protein alpha subunit (RGA1) to be an important node in the PPI network (Appendix A). Our earlier studies revealed the role of heterotrimeric G-protein subunit in the regulation of N-associated genes in maize [69] and rice [70]. Genetic analyses also revealed a gene for G-protein gamma subunit in the QTL for NUE in rice [71]. Our earlier transcriptomic works on G-protein alpha subunit mutant in Arabidopsis (*gpa1*) and rice (*rga1*/*d1*) suggested its role in the regulation of N-signaling and -response in plants [72,73]. Further, it has been shown that d1 mutant also regulates NUE in rice [74] (Jangam et al., submitted). Our current findings based on PPI networks carry forward these findings by providing mechanistic insights into the underlying genes/processes for informed interventions.

We propose a hypothetical model on the possible role of urea-responsive DEGs in NUE (Figure 7), which broadly classified into two categories, viz., those DEGs and associated processes/pathways enriched in high-NUE genotype, while others enriched in low- NUE genotype. We observed that in response to urea, distinct set of transporters and TFs are differentially enriched in contrasting genotype. For example, *OsPIN1d* (Auxin Efflux Carrier (AEC) Family: Os12g0133800) was not detected in Nidhi, whereas it was up-regulated in Panvel1 (Appendix A). It has been shown that rice plant knocked out for *OsPIN1d* along with *OsPIN1c* gives decreased plant height, tiller number and severely disrupted panicle development [28]. Our data shows its upregulation in Panvel1 could be linked to its high NUE over Nidhi. It has been shown that *OsGRF4* transcription factor promotes hormone-dependent NUE enhancement in rice [75]. Most of the DEGs from GRAS family TFs were up-regulated in Panvel1, whereas they were not detected in Nidhi under low urea (Appendix A). This suggests their role for efficient NUE in Panvel1. Finally, our field evaluation of these two rice genotypes, namely Nidhi and Panvel1, contrasting for NUE, clearly reveals that high yield and nitrogen use efficiency of genotype Panvel1 over the Nidhi is due to better nitrogen uptake, better N-transport and better N-utilization (Figure 7).

The findings of this study are based on whole plant microarray and RT-qPCR analyses that were not designed to address stage-specific or tissue-specific patterns of expression/regulation. However, we showed elsewhere [17] that the N-responsive phenotypic differences established during early stages of rice growth are retained until the active tillering stage. We also demonstrated there by RT-PCR that the pattern of expression of several DEGs in 30-days-old excised rice leaves (treated with nitrate in vitro) matched with that observed in the leaves of 20-days-old potted plants (treated with nitrate in vivo). Nevertheless, it is advisable to re-check tissue/stage-specific expression of the gene/process for informed interventions. Overall, our comparative transcriptomic analysis of urea response in rice genotypes with contrasting NUE provides important mechanistic insights and also offers novel candidate genes for crop improvement.

## 4. Materials and Methods

### 4.1. Plant Material, Growth Conditions, N-Treatments and Physiological Measurements

Two genotypes of rice (*Oryza sativa* ssp. Indica), namely Nidhi and Panvel1, were chosen based on contrasting germination, yield and NUE [10,11]. Their seeds of modal weight were surface-sterilized and grown in pots containing nutrient-depleted soil, exactly as described [10,55]. The pots were saturated with modified Arnon-Hoagland medium containing urea as the sole N source at 15 mM (normal) or 1.5 mM (low) concentration as control and test conditions, as described earlier [11]. For this purpose five independent pots, each sown with two seeds, were used to grow the plants and harvested on the 21st day, and the same healthier plant was used for all the measurements. The pots were replenished with media to saturation every few days and plants were grown for 21 days in the green house at 28 °C and 70% relative humidity with 270 μmol m^−2^ s^−1^ light intensity and 12/12 hr photoperiod. These plants were used to measure photosynthesis, stomatal conductance and transpiration rate using LI-6400XT Portable Photosynthesis System (LI-COR Biosciences, Lincoln, NE, USA).

### 4.2. Microarray Analyses

Harvested 21-day whole plants from three independent biological replicates were used to extract total RNAs and used for microarray analyses under MIAME compliant conditions exactly as described earlier [16]. The raw data and processed data were deposited in the NCBI-GEO database (GSE140257). A GO based functional annotation of differentially expressed genes (DEGs) was performed using EXPath 2.0. (http://EXPath.itps.ncku.edu.tw accessed on 1 January 2021). MS Excel was used for filtering of the data and Student’s *t*-test was used for statistical significance. Venn diagrams were made using Venny 2.1 tool (https://bioinfogp.cnb.csic.es/tools/venny/ accessed 1 September 2021).

PlantPAN3 (http://plantpan3.itps.ncku.edu.tw/ accessed on 1 September 2021) was used to retrieved TFs encoded by DEGs of both Nidhi and Panvel1. A 2 kb promoter regions upstream to the transcriptional initiation sites were downloaded from RAPDB and used for identification of transcription factor binding sites (TFBS) by RSAT tool (http://plants.rsat.eu accessed on 1 September 2021). Transporters encoded by DEGs were retrieved from the Rice Transporters Database (https://ricephylogenomics.ucdavis.edu/transporter/ accessed on 15 January 2021) and Transport DB 2.0 (http://www.membranetransport.org/transportDB2/index.html accessed on 15 January 2021). The Plant miRNA database was used to retrieve the miRNAs that target NUE-related genes (PMRD: http:// bioinformatics.cau.edu.cn/PMRD/ accessed on 5 August 2021). The database Plant PTM Viewer was used to find the products of DEGs associated with post-translational modifications (PTM) (https://www.psb.ugent.be/webtools/ptm-viewer/experiment.php accessed on 1 September 2021). N-responsive yield related genes that were defined as NUE genes and identified as described by Kumari et al. [14], and they were co-localized to NUE-QTLs for filtering the important NUE-candidates.

### 4.3. RT-qPCR Validation of Urea-Responsive Expression of DEGs

Twenty-one days old whole plants grown in normal and low urea treatments (15 mM as control and 1.5 mM as test) were used to isolate total RNAs. cDNAs were synthesized using 3 µg each of the total RNA and PrimeScript 1st strand cDNA synthesis kit (Takara, Kusatsu, Shiga, Japan). Exon spanning primes were made using the Quant Prime tool (https://quantprime.mpimp-golm.mpg.de/?page=about accessed on 1 April 2021) to eliminate the chances of amplifying the genomic DNA. RT-qPCR reactions were set up in an Agilent Aria-Mx Real-Time PCR exactly as described earlier [16]. The relative changes in gene expression were quantified by 2^−△△CT^ method [76] using actin gene (BGIOSGA013463) as an internal control. Melting curve analyses of the amplicons were used to determine the specificity of RT-qPCR reactions. The data were statistically analyzed by an unpaired *t*-test using MS Excel software.

### 4.4. Field Experiments for Agronomic, Physiological and NUE

In order to test dose-specific urea responses on agronomical and physiological traits and NUE, contrasting rice genotypes Nidhi and Panvel were evaluated under field conditions at the ICAR-National Rice Research Institute, Cuttack, Odisha, India during tye Kharif (summer) season of 2021. Seeds were sown on 5 July 2021 and one-month-old seedlings were transplanted on 10 August 2021 in a split-plot design. The plot size was 5.4 m^2^ (10 rows × 18 hills) with N levels as the main plot and varieties as the subplot in two replicates. The crop geometry was 20 cm between rows and 15 cm between plants. Urea was the sole source of applied N at the rate of 50 (N50) and 100 (N100) kg N/ha in three splits (1/2 at basal, 1/4 at vegetative and 1/4 at flowering stage), with a control of no added N (N0). The same plots were used for N0, N50 and N100 for both the genotypes. The characteristics of the field soil before and after the experiment, along with the total carbon and nitrogen contents of plants, are provided as supplementary methods.

The genotypes were harvested on 15 November 2021 at physiological maturity and various yield parameters, viz., yield per plant, grain yield per hectare and 1000 grain weight were recorded. NUE indices were calculated as per their standard definitions as follows: partial factor productivity (PFP, kg grain/kg N added; grain yield/N applied), uptake efficiency UpE = N uptake (NUpt)/N supplied (Ns), nitrogen transport efficiency (NTE = total N transport into the above ground parts/total N in to the whole plants or N-uptake by plant), utilization efficiency = grain weight/total N in plant and fertilizer use efficiency = (fertilizer uptake/fertilizer applied) × 100 were measured at maturity stage only.

## 5. Conclusions

Transcriptomic analysis of urea response in two rice genotypes contrasting for NUE revealed differential involvement of biological processes, transporters, TFs and their networks, miRNAs, post-translational modifications and NUE-candidates co-localized onto NUE QTLs in a genotype-dependent manner. Comparative analyses between urea- and earlier nitrate-responsive transcriptomes revealed N form specific differential regulation of transporters and transcriptional factors, which may account for contrasting NUE in rice genotypes. The high-NUE genotype, Panvel1 showed a better photosynthetic rate and water use efficiency in potted plants under greenhouse conditions, relative to Nidhi. A field evaluation of the contrasting genotypes under different doses of urea revealed better performance of Panvel1 in different agronomic parameters including grain yield, transport, uptake efficiencies and NUE.

## Figures and Tables

**Figure 1 ijms-24-06080-f001:**
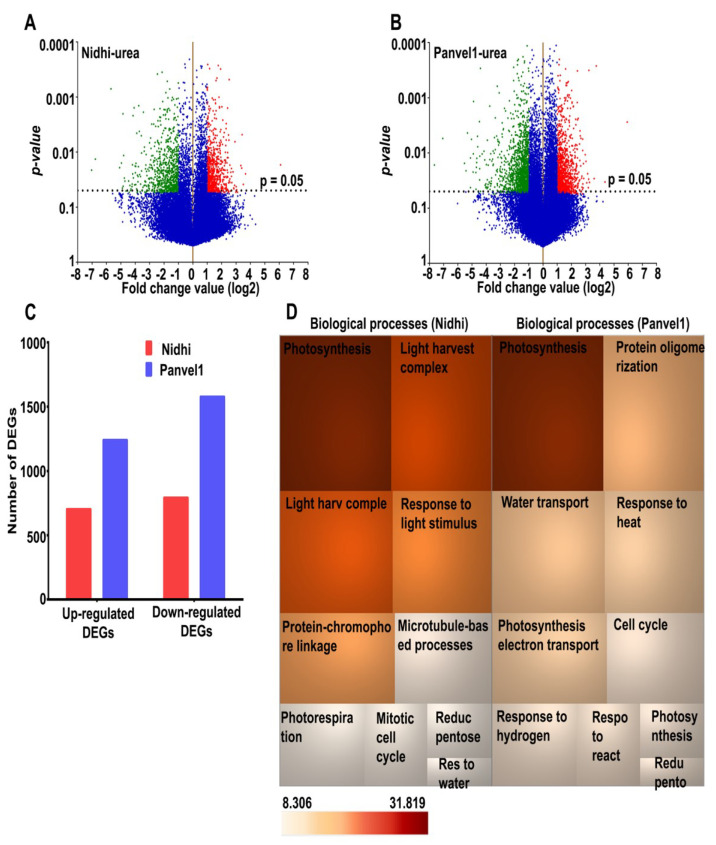
Transcriptomic analyses of low urea response in contrasting NUE genotypes. Nidhi and Panvel1 indica rice genotypes were grown in nutrient depleted soil in greenhouse under normal (15 mM) and low (1.5 mM) urea conditions [10]. Urea-responsive differentially expressed transcripts are shown as volcano plots for Nidhi (**A**) and Panvel1 (**B**). Each dot on the plot represents the transcript and horizontal dashed line corresponds to *p* value cut−off (*p* = 0.05). Up-regulated transcripts are shown as red scattered dots, whereas green scattered dots represent down-regulated transcripts under low−urea treatment. (**C**) Bar graph represents the up− or down−regulated genes detected in Nidhi and Panvel1. (**D**) TreeMap shows top 10 statistically significant biological processes (*p* < 0.05) observed in Nidhi and Panvel1. The colour of the box is according to the *p* value (−log_2_).

**Figure 2 ijms-24-06080-f002:**
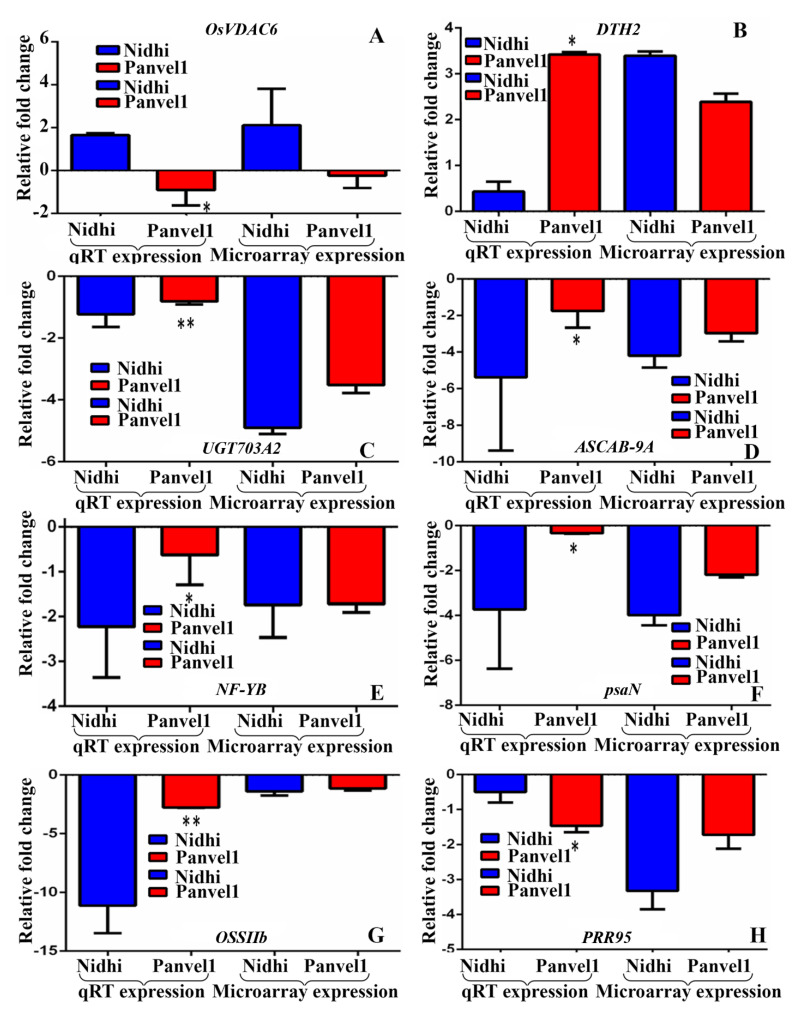
Validation of expression profile of urea-responsive genes by RT−qPCR. Relative change in the gene expression was calculated by comparative Ct value method and actin gene was used for data normalization. The control values were taken as zero and the test values are shown as average of three technical and two independent biological replicates (+SE). Each sub−figure compares gene expression of RT−qPCR and microarray for Nidhi versus Panvel1 for gene *OsVDAC6* (**A**), *DTH2* (**B**), *UGT703A2* (**C**), *ASCAB-9A* (**D**), *NF-YB* (**E**), *psaN* (**F**), *OSSIIb* (**G**) and *PRR95* (**H**). Test of significance between bars has been shown on RT−qPCR data of Nidhi versus Panvel1. * *p*  <  0.05, ** *p*  <  0.01.

**Figure 3 ijms-24-06080-f003:**
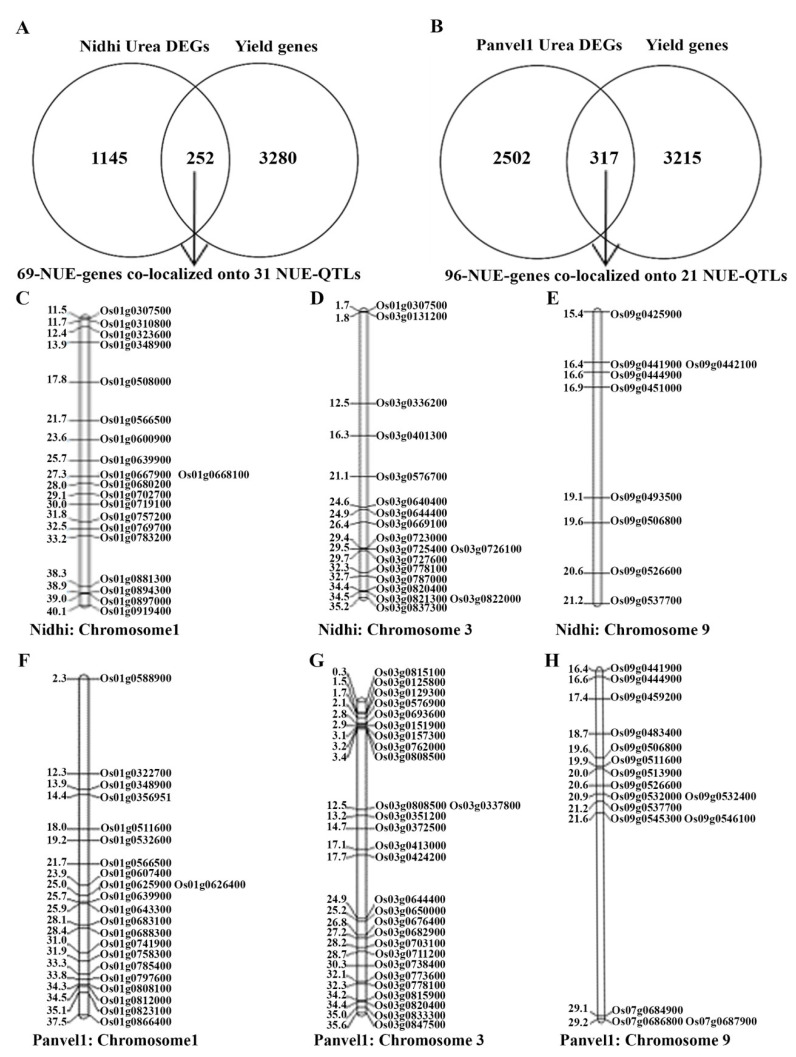
Venn selection for NUE genes (N-responsive and yield-related genes) in (**A**) Nidhi urea and (**B**) Panvel1 urea. Representative figures for physical location of NUE candidates located on chromosomes 1, 3 and 9 in (**C**–**E**) Nidhi urea and (**F**–**H**) Panvel1 urea. Gene ID is given on the right side and the physical location of genes is given on the left side of the map (in Mb).

**Figure 4 ijms-24-06080-f004:**
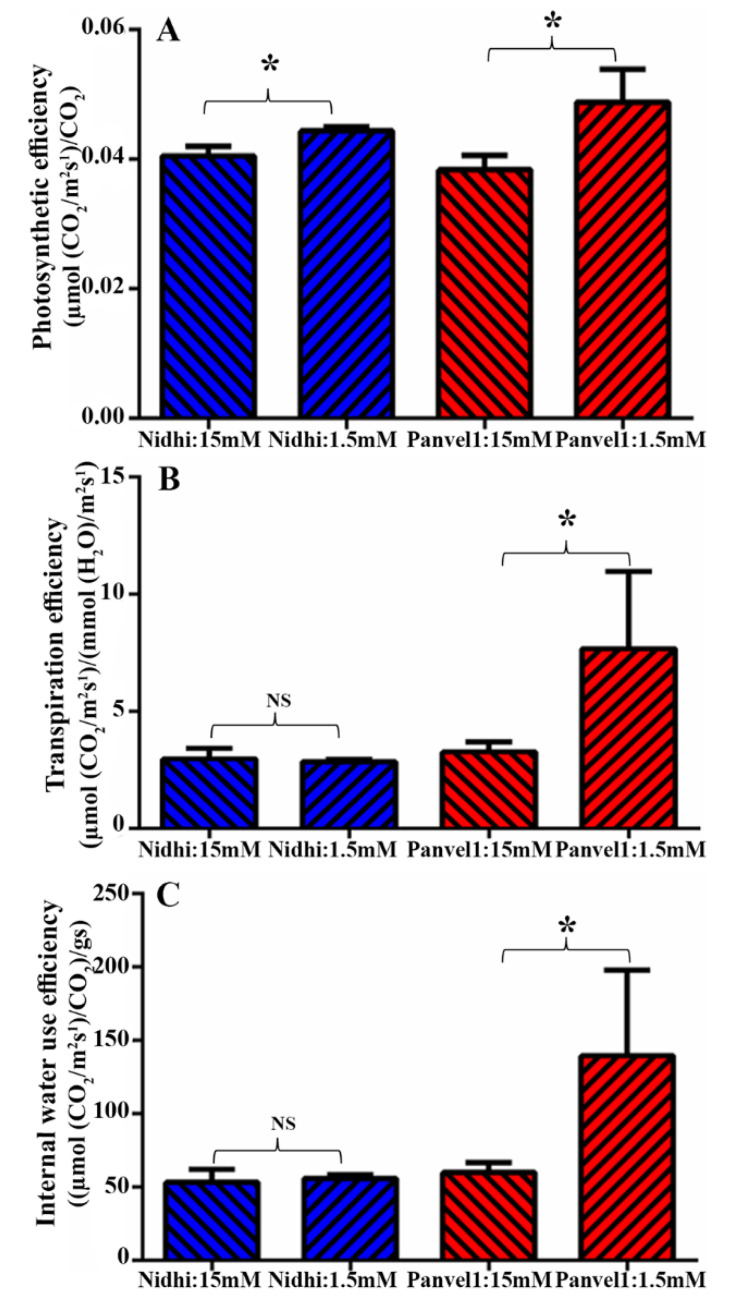
Validation of efficiencies derived associated with the biological processes. Validation was carried out using Licor instrument 6400XT (LI-COR, Lincoln, NE, USA) on 21−days-old grown plants. Plants were grown in nutrient-depleted soil and fertilized with Arnon Hoagland medium having urea as sole source of N with 15 mM N concentration as control, while 1.5 mM was used as test. Measurement was carried out in four biological replicates. Efficiencies were derived from the standard formulas as described here. (**A**) Photosynthetic efficiency was measured in terms of µ mol CO_2_m^−2^s^−1^/µ mol l^−1^, (**B**) transpiration efficiency was measured in terms of µ mol CO_2_m^−2^s^−1^/mmol (H_2_O)/m^−2^s^−1^ and (**C**) internal water use efficiency was measured in terms of µ mol CO_2_ m^−2^s^−1^/mol(H_2_O) m^−2^s^−1^). The test of significance has been shown as star (* *p* < 0.05) between the bars of low urea and normal urea, while NS represents non-significance.

**Figure 5 ijms-24-06080-f005:**
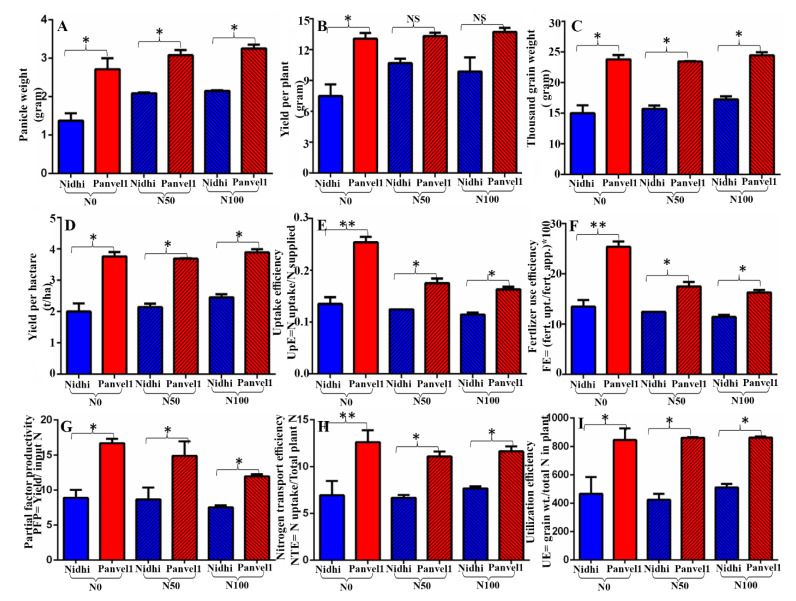
Field validation of the performance of rice genotypes. Field evaluation of the genotypes Nidhi and Panvel1 was conducted at National Rice Research Institute (NRRI-ICAR) Cuttack, Odisha, India in N0, N50 and N100 N kg of added urea/ha. Mean data of the effect of urea dose on the genotypes are shown for panicle weight, yield per panicle, 1000 grain weight, grain yield per hectare, uptake efficiency, fertilizer use efficiency, partial factor productivity, nitrogen transport efficiency and utilization efficiency. The significance levels are shown between the bars of genotype Nidhi and Panvel1 for each of the urea dose being compared (* *p*  <  0.05, ** *p*  <  0.01 and NS denotes non-significant).

**Figure 6 ijms-24-06080-f006:**
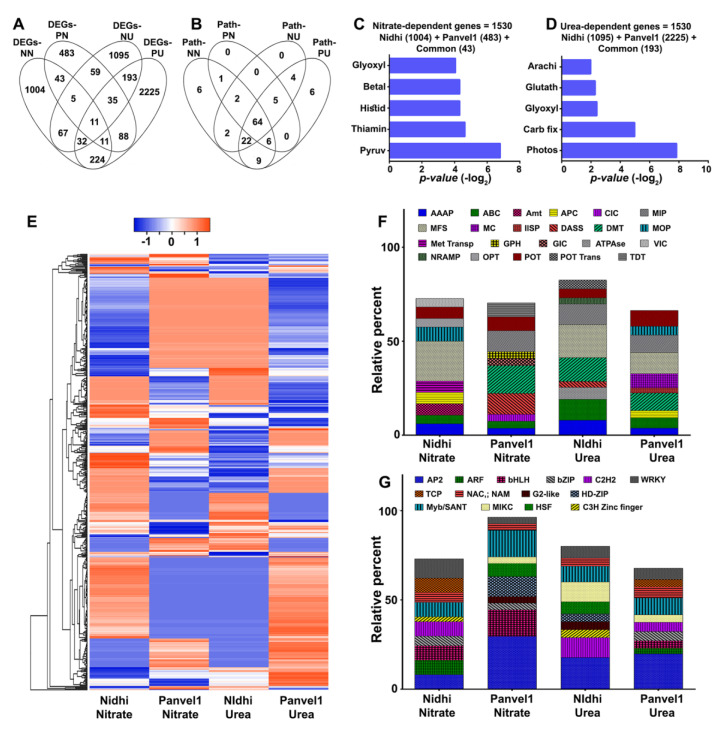
Common and specific nitrate− and urea−responsive genes and pathways in Nidhi and Panvel1. Venn diagrams represent the specific and overlap of DEGs (**A**) and all the assigned pathways (**B**) in low nitrate versus low urea responses in both the genotypes. Combined N−responsive genes and their top enriched pathways annotations are shown for nitrate (**C**) and urea (**D**). (**E**) Hierarchical clustering of common N−responsive genes from Nidhi shows distinct pattern in Panvel1. Each column represents a single N (nitrate or urea) treatment in a particular genotype (Nidhi or Panvel1), whereas each row represents a DEG. Stacked bar graph represents the relative percentage of top 10 enriched transporters (**F**) and transcription factors (**G**) responsive to nitrate [16] and urea (current study). The X−axis represents the nitrate or urea treatment in Nidhi or Panvel1. Details of genes and pathways are given in Appendix A. DEGs−NN, DEGs−Nidhi nitrate; DEGs−PN, DEGs−Panvel1 nitrate; DEGs−NU, DEGs−Nidhi urea; DEGs−PU, DEGs−Panvel1 urea; Path−NN, Pathways−Nidhi nitrate; Path−PN, Pathways−Panvel1 nitrate; Path−NU, Pathways−Nidhi urea; Path−PU, Pathways−Panvel1 urea.

**Figure 7 ijms-24-06080-f007:**
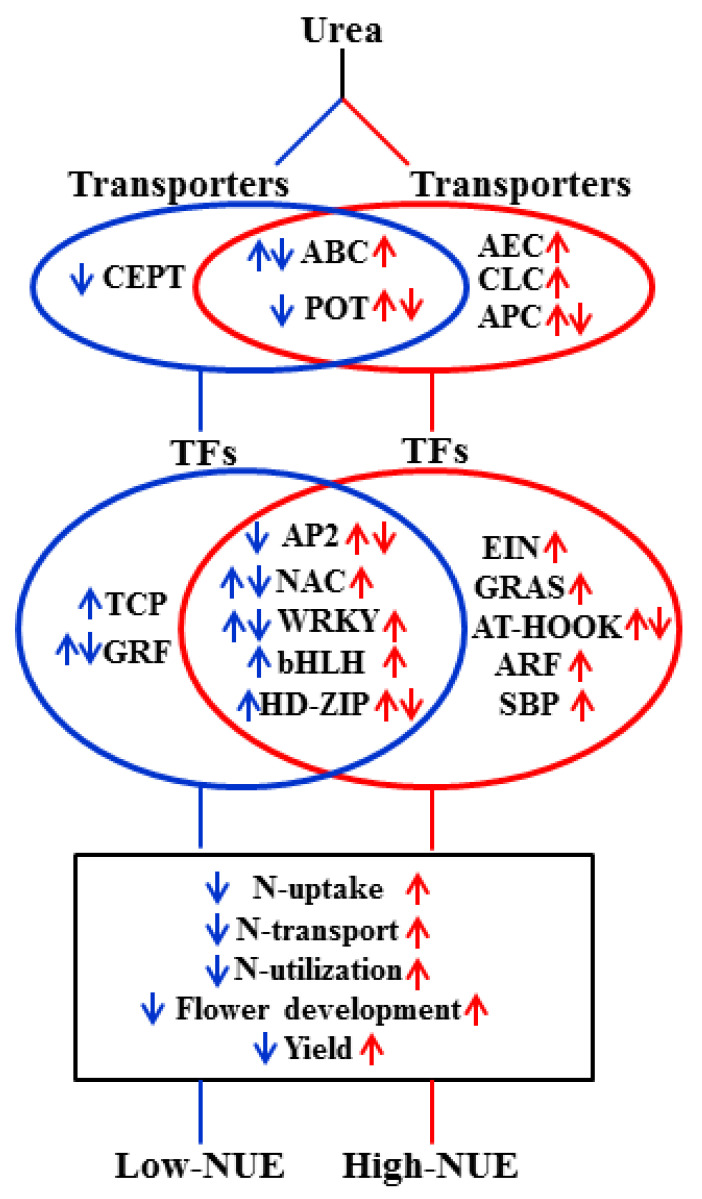
Hypothetical model depicting the important classes of genes and associated pathways differentially regulated in low−(Nidhi) and high−(Panvel1) NUE genotypes. Blue and red colour denote the differential regulation of gene classes/pathways in low− and high−NUE genotypes, respectively. Up− and down−ward arrows represent the upregulation and downregulation, respectively, in a particular class/pathways. Both the arrows together represent mixed regulation of gene classes/pathways. TFs, transcription factors.

**Table 1 ijms-24-06080-t001:** Details of the DEGs associated with various functional classes.

Functional Categories	Total Number of DEGs		Number of Up-Regulated DEGs		Number of Down-Regulated DEGs	
	Nidhi	Panvel1	Nidhi	Panvel1	Nidhi	Panvel1
Transporters	63	107	33	57	30	50
Transcription factors	45	96	20	74	25	22
miRNA	45	57	23	40	22	17

## Data Availability

GEO accession number: GSE 140257.

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
