# Peer review of "Genome-Wide Urea Response in Rice Genotypes Contrasting for Nitrogen Use Efficiency"

_ijms, 2023, doi:10.3390/ijms24076080_

Round 1
Reviewer 1 Report
In this manuscript, the authors set out to Genome wide Urea Response in Rice Genotypes Contrasting for Nitrogen Use Efficiency chromi, and help us understand its role at the plant and soil interface. This is an interesting topic and merits a research article.
Author Response
Reviewer #1
Comments and Suggestions for Authors:
In this manuscript, the authors set out to Genome wide Urea Response in Rice Genotypes Contrasting for Nitrogen Use Efficiency, and help us understand its role at the plant and soil interface. This is an interesting topic and merits a research article.
Reply: Thanks for your comments and sparing your valuable time to look into our manuscript. We have thoroughly revised the manuscript including text, grammar, and typos as suggested by reviewer.

Reviewer 2 Report
This work shows an interesting description of the transcriptome for N uptake in rice. The results are important to help to elucidate the response of this crop to N-urea fertilization and the associated gene.
In order to improve the Results section (is very large, repetitive, and mentions too much the Supplementary tables), I recommend adding a table for the main results in order to summarize the data obtained regarding the up -dow regulation of gene expression and TFs involved.
Some minor details to fix:
Line 15: Define DEGs acronym the first time in Abstract.
Line 35 Define the NUE acronym the first time in text.
Lines 38-40, this phrase is not clear, please define "biological interventions" or provide an example.
Line 81: "enriched" maybe is not the best word in this context.
Line 391: "huge" is not appropriate adjective, describe quantitatively.
Figure legend 1: Lines 856-857, The method description should be in the Materials & Methods Section.
Figure legend 2: Please add the meaning of * and ** here and following figures (e.g. Fig 6)
Author Response
Reviewer #2
Comments and Suggestions for Authors:
This work shows an interesting description of the transcriptome for N uptake in rice. The results are important to help to elucidate the response of this crop to N-urea fertilization and the associated gene.
Reply: Thanks for your comments and sparing your valuable time to look into our manuscript.
Query 1. In order to improve the Results section (is very large, repetitive, and mentions too much the Supplementary tables), I recommend adding a table for the main results in order to summarize the data obtained regarding the up -down regulation of gene expression and TFs involved.
Reply: Thanks. As suggested, a Table summarizes major findings related to up- and down-regulated DEGs has been added and repetitions minimized in the revised manuscript.
Some minor details to fix:
Query 2. Line 15: Define DEGs acronym the first time in Abstract.
Reply: Thanks. As suggested, the expansion for the acronym ‘DEGs’ has been added in the text at the first place used in the revised manuscript.
Query 3. Line 35 Define the NUE acronym the first time in text.
Reply: Thanks. As suggested, the expansion for the acronym ‘NUE’ has been added in the text at the first place used in the revised manuscript.
Query 4. Lines 38-40, this phrase is not clear, please define "biological interventions" or provide an example.
Reply: Thanks. Sentence has been edited in the revised manuscript.
Query 5. Line 81: "enriched" maybe is not the best word in this context.
Reply: Thanks. Sentence has been edited in the revised manuscript.
Query 6. Line 391: "huge" is not appropriate adjective, describe quantitatively.
Reply: Thanks. Sentence has been edited in the revised manuscript.
Query 7. Figure legend 1: Lines 856-857, The method description should be in the Materials & Methods Section.
Reply: Thanks. To avoid the repetition, the sentence has been deleted in the revised manuscript.
Query 8. Figure legend 2: Please add the meaning of * and ** here and following figures (e.g. Fig 6)
Reply: Thanks. As suggested, all the figure legends were rechecked and the meaning of “*”/“**”and other related statistical details have been added in the revised manuscript.
